# The Characterization of Cardiac Explants Reveals Unique Fibrosis Patterns and a Predominance of CD8+ T Cell Subpopulations in Patients with Chronic Chagas Cardiomyopathy

**DOI:** 10.3390/pathogens11121402

**Published:** 2022-11-23

**Authors:** Martha Lucía Díaz, Fredy A. Delgado, Ruth A Martínez, Mayra Alejandra Jaimes, Luis Eduardo Echeverría, Sergio Alejandro Gómez-Ochoa, Julio César Mantilla-Hernández, Clara Isabel González

**Affiliations:** 1Immunology and Molecular Epidemiology Group, School of Microbiology, Universidad Industrial de Santander, Bucaramanga 680002, Colombia; 2Heart Failure Clinic, Fundación Cardiovascular de Colombia, Floridablanca 681004, Colombia; 3Department of Pathology, Universidad Industrial de Santander, Bucaramanga 680002, Colombia

**Keywords:** chagas disease, chronic chagas cardiomyopathy, histopathology, immunophenotyping

## Abstract

Aim: The present study aimed to characterize the histopathological findings and the phenotype of inflammatory cells in the myocardial tissue of patients with end-stage heart failure (ESHF) secondary to CCC in comparison with ESHF secondary to non-Chagas cardiomyopathies (NCC). Methods: A total of 32 explanted hearts were collected from transplanted patients between 2014 and 2017. Of these, 21 were classified as CCC and 11 as other NCC. A macroscopic analysis followed by a microscopic analysis were performed. Finally, the phenotypes of the inflammatory infiltrates were characterized using flow cytometry. Results: Microscopic analysis revealed more extensive fibrotic involvement in patients with CCC, with more frequent foci of fibrosis, collagen deposits, and degeneration of myocardial fibers, in addition to identifying foci of inflammatory infiltrate of greater magnitude. Finally, cell phenotyping identified more memory T cells, mainly CD8+CD45RO+ T cells, and fewer transitioning T cells (CD45RA+/CD45RO+) in patients with CCC compared with the NCC group. Conclusions: CCC represents a unique form of myocardial involvement characterized by abundant inflammatory infiltrates, severe interstitial fibrosis, extensive collagen deposits, and marked cardiomyocyte degeneration. The structural myocardial changes observed in late-stage Chagas cardiomyopathy appear to be closely related to the presence of cardiac fibrosis and the colocalization of collagen fibers and inflammatory cells, a finding that serves as a basis for the generation of new hypotheses aimed at better understanding the role of inflammation and fibrogenesis in the progression of CCC. Finally, the predominance of memory T cells in CCC compared with NCC hearts highlights the critical role of the parasite-specific lymphocytic response in the course of the infection.

## 1. Introduction

Chagas disease (CD) is a disease endemic to Latin America caused by the protozoan parasite *Trypanosoma cruzi* (*T. cruzi*). However, due to migratory phenomena, CD has become a worldwide public health problem, being recognized as the parasitic disease with the highest attributable disease burden in the world [1,2,3]. Currently, it is estimated that more than 6 million people are infected with *T. cruzi*, and 65 million are at risk of infection, with an increasing trend in disease prevalence in North America and Europe [4,5]. 

CD is characterized by an initial acute phase followed by a chronic phase in which 20% to 30% of patients will present target organ involvement [6]. When the heart is the affected organ, it is known as chronic Chagas cardiomyopathy (CCC), which represents the most severe form of organ involvement in CD [7]. This entity is characterized by the presence of severe conduction disorders, mainly atrial fibrillation and ventricular arrhythmias, as well as the development of structural alterations of the left ventricle such as aneurysms and functional alterations such as contractility disorders, among others [3]. All this positions CCC as one of the etiologies of heart failure with the highest morbidity and mortality, which, added to its high prevalence in endemic countries, confers a high burden of disease [4,8]. Unfortunately, there is no effective treatment for this condition once myocardial involvement has been established [6].

The pathogenic mechanisms responsible for the development of myocardial involvement during the chronic phase of CD have not been fully elucidated [3]. An important pathological feature is the presence of an intense inflammatory infiltrate in the myocardium, which seems to act as an effector mechanism of cardiac damage [7]. Some studies performed of biopsies and tissues have highlighted the presence of infiltrates consisting of macrophages and mononuclear cells; however, the relationship between their presence and the development of myocardial injury is still not fully understood [9,10,11]. Furthermore, so far, no comparative characterization has been made of the distribution and intensity of histological changes in the different cardiac chambers, especially cardiomyocyte hypertrophy and fibrosis, taking into account the importance of these histopathological changes in the course of the disease. Moreover, the cellular phenotype of the inflammatory infiltrate has been poorly studied in tissues of patients with advanced cardiomyopathy, which significantly limits the understanding of the pathophysiology of the disease [9]. Therefore, the present study aimed to perform a comparative analysis of the histopathological features and phenotype of the inflammatory infiltrate in patients with CCC compared with those with heart failure of other etiologies.

## 2. Materials and Methods

### 2.1. Study Design and Setting 

A total of 32 explanted hearts were collected from transplanted patients between 2014 and 2017. Of these, 21 were classified as CCC and 11 as non-Chagas cardiomyopathy (NCC). Patients were classified in the CCC group when there were two positive serological tests detecting *T. cruzi* infection (ELISA and IFA) and echocardiographic or electrocardiographic abnormalities (left anterior fascicular block, right bundle branch block, atrioventricular blocks, ventricular premature beats, atrial fibrillation or flutter, bradycardia ≤ 50 beats/min, and/or Echo findings suggestive of myocardial impairment) consistent with chronic Chagas cardiomyopathy as evaluated by a cardiologist. All hearts were obtained from the cardiovascular surgery service of the Fundación Cardiovascular de Colombia (FCV). Relevant information was extracted from the institutional medical record at the time of tissue collection, including sociodemographic, clinical, and echocardiographic data. Echocardiograms were routinely performed according to the FCV management algorithms for patients listed for heart transplantation.

### 2.2. Macroscopic Analysis

Each heart was systematically weighed, inspected, measured, and examined for pathological changes in the left ventricle (LV), right ventricle (RV), ventricular septum (VS), left atria (LA) right atria (RA), pericardium, endocardium, and coronary arteries. Other features evaluated included chamber enlargement/dilation; myocardial and epicardial scarring; and the presence of mural thrombi, aneurysms, and epicardial plaques. 

### 2.3. Histopathological Analysis

Transverse sections of 2 cm were made from the cardiac base, obtaining fragments for subsequent microscopic study. Multiple samples of myocardium were fixed in formaldehyde, embedded in paraffin, and stained with hematoxylin and eosin (H&E). Histological specimens were evaluated for the presence of fibrosis, necrosis, and variation in nuclei size. Paraffin-embedded sections of the mid-ventricle were stained with a collagen stain (Masson’s trichrome) to evidence fibrosis.

Histopathological analyses included quantification of fibrosis and hypertrophy to assess the extent of reactive (interstitial/perivascular) and replacement fibrosis, hypertrophy, and myocytolysis. Briefly, each myocardial specimen was histopathologically scored with a grade of 0 = negative; 1 = mild (occasional foci/scarring); 2 = moderate (multiple foci/scarring); or 3 = severe (extensive fibrotic involvement). Hypertrophy was identified as an increase in cardiomyocyte (CMC) diameter, as well as an increase in the size and hyperchromasia of their nuclei. The degree of CMC hypertrophy was classified as 0 = absent, 1 = mild (occasional foci), 2 = moderate (multiple foci), or 3 = severe (extensive/diffuse hypertrophy). The severity of myocarditis was analyzed using conventional histology, including the degree of inflammatory cell infiltration in each region, which was quantified by counting the absolute number of infiltrating inflammatory cells in a 100× magnification field and examining their distribution. The severity and extent of fibrosis, hypertrophy, and myocytolysis were examined in a blinded fashion, as the pathologist in charge of this analysis had no prior knowledge of the cause of heart failure, age, or other clinicopathological patient information.

In addition to the previously described classification system for the pathological assessment of HF, collateral myocytolysis was defined as the presence of CMC with some stage of vacuolization and myofibril loss; its severity was classified as 0 = absent, 1 = mild, 2 = moderate, or 3 = severe. The LV, RV, VS, LA, and RA of each heart received a total score for fibrosis, hypertrophy, and myocytolysis, with a mean number of histological specimens/sections per area of 4, 2, 1, 1, and 1, respectively.

### 2.4. Immunohistochemical Analysis

We performed a descriptive immunohistochemistry analysis of samples from patients in the CCC group with the aim of generating hypotheses about the expression of different molecules potentially important for the pathophysiology of Chagas disease. For this purpose, tissue sections were collected on poly-L-lysine coated slides and dried in a convection oven at 90 °C for 24 h. Then, the classical protocol for sections was followed: deparaffinization followed by hydration. To unmask the antigen, the slides were boiled in a sodium citrate solution at pH 6 for 1 h. After boiling and cooling, the slides were washed in distilled water for 15 min. Endogenous peroxidase blocking was performed by incubating the slides in 3% hydrogen peroxide for 30 min at room temperature, followed by washing in distilled water for 10 min and washing in 1% phosphate-buffered saline (PBS) solution for five minutes. Nonspecific sites were then blocked using 2% skim milk for 30 min. The sections were then incubated with antibodies against type IV collagen (ab 34710) and XIVA1 (NBP1-86877) using a 1:50 dilution for 18 h (overnight) in a refrigerator at 4 °C. Secondary detection was performed with a peroxidase-conjugated goat anti-rabbit secondary antibody at 1:10,000 for 45 min at room temperature. The signal was detected using 3,3′ Diaminobenzidine (DAB) (Abcam), and the reaction was quenched in 1% PBS.

### 2.5. PCR Analysis

For amplification of the 188 bp fragment of the *T. cruzi* nuclear DNA repeat sequences (nDNA), primers TCZ-I (5′-CGA GTC CTT GCC CAC ACG GGT GCT-3′) and TCZ-II (5′-CCT CCA AGC AGC GGA TAG TTC AGG-3′) were used. Myocardial tissue samples from the left ventricular free wall were used for this purpose. Amplifications were performed in a final volume of 20 μL containing 10 mM Tris HCl pH 9.0, 1.5 mM MgCl_2_, 50 mM KCl, 0.1% Triton X-100, 200 mM of each of the dNTPs, 0.5 μM of each primer, 0.6 U of Taq DNA polymerase (Corpogen, Bogotá, Colombia) and 1 μL of tissue DNA or 1 ng of parasite DNA (controls). Amplification was performed for 40 cycles with denaturation at 94 °C for 20 s, annealing at 60 °C for 10 s, and extension at 72 °C for 30 s, followed by a final extension at 72 °C for 7 min. A 20 μL volume of the amplified products was subjected to 2% agarose gel electrophoresis in 1× TAE buffer for 90 min at 90 V, visualized by Safe View-classic (abm) staining and photographed with the ChemiDoc^TM^ MP Imaging System version 6.0 (Bio-Rad, Hercules, CA, USA). 

### 2.6. Flow Cytometry

#### 2.6.1. Preparation of Cell Suspension

To determine the composition of the inflammatory infiltrate, 4 g of tissue were taken from the posterior basal wall of the left ventricle in nine explanted hearts (four randomly selected samples from the CCC group and five from the NCC group). The tissue was cut into small fragments (<1 mm^3^), and the fragments were deposited in a calcium-free solution (20 mM glucose, 10 mM KCl, 1.2 mM KH_2_PO_4_, 5 mM MgSO_4_, 5 mM MOPS, 100 mM NaCl, 50 mM taurine, pH = 7.0 adjusted with 1 M NaOH) at 4 °C to ensure cell viability. Next, the tissue was allowed to decant for 10 min, and then the supernatant was removed, and 40 mL of a calcium-free solution supplemented with 1 mg collagenase (286 U/mL) (Sigma-Aldrich, St. Louis, MO, USA); then 2.5% penicillin/streptomycin was added and shaken for 10 min at 37 °C. After this time, 80 μL of 10 mM CaCl_2_ solution was added to obtain the final concentration of 20 μM Ca. The digested material was homogenized by gentle trituration with a plunger before being filtered. Filtration was performed by passing the sample through a 70 μm nylon mesh filter into a 50 mL falcon tube. The collected suspension was subjected to density gradient centrifugation using Ficoll-Paque^TM^ (GE Healthcare Bio-Sciences, Pittsburgh, PA, USA) at a 3:1 ratio. After centrifugation at 2000 rpm for 40 min at room temperature, the layer containing the mononuclear cells was harvested, and the cells were washed twice with PBS supplemented with 5% fetal bovine serum (FBS) and then deposited on the flow cytometer.

#### 2.6.2. Antibodies

Monoclonal antibodies (MAbs) recognizing T lymphocyte-associated (CD3, CD4, and CD8) and B lymphocyte-associated (CD10 and CD19) surface antigens labeled with fluorescein isothiocyanate (FITC) were used (Ortho Diagnostic Systems. Raritan, NJ, USA).

#### 2.6.3. Phenotype Evaluation

To perform the surface marker study, first, the absolute counts per µL of fresh leukocytes were determined by counting live (trypan blue-negative) cells in 200 μL of staining buffer in a Newbauer chamber. The mean and SD for live cell recoveries from individual cardiac tissues were (7.0 ± 2.0) × 10^5^/heart. Then, the harvested cells were incubated with different monoclonal antibodies in a dark room for 30 min at room temperature. All antibodies were directly labeled. Furthermore, a final volume of 100 μL per panel was injected on the BD FACSCanto flow cytometer (Becton Dickinson, San Jose, CA, USA) and analyzed using the Infinicyt software version 2.0 (Cytognos, Salamanca, Spain). Four different panels of antibodies were used to perform the phenotypic characterization of mononuclear cells according to the Euroflow multicolor panels (Table 1).

#### 2.6.4. Evaluation of the Intracellular FoxP3 by Flow Cytometry

The IntraPrepTM Permeabilization Reagent kit (Beckman Coulter, Brea, CA, USA) was used for the evaluation of intracellular FoxP3. Briefly, the mononuclear cells recovered from the tissue were incubated with the respective surface markers under the conditions mentioned above, then fixed with 40 μL of reagent 1 (5% (*v/v*) formaldehyde), mixed vigorously with a vortex, and then incubated for five minutes in darkness. Then 1 mL of lysis buffer was added and incubated again for five minutes in the dark at room temperature. Subsequently, it was centrifuged at 2000 rpm for 5 min; the supernatant was discarded; and 40 μL of reagent 2 (0.1% (*w*/*v*) NaN_3_) was added, vortexed, and incubated for an additional five minutes. Finally, the intracellular marker FoxP3 (10 μL) was added and mixed with vortex and incubated for 20 min in the dark at room temperature.

### 2.7. Statistical Analysis and Ethical Considerations

Qualitative variables were described using absolute and relative frequencies, while normally distributed quantitative variables were described using means and standard deviations. On the other hand, quantitative variables that failed to prove normality were described as medians and interquartile ranges (IQR). Differences between the two groups were assessed using Pearson’s chi-squared and Fisher’s exact tests for categorical variables and Mann–Whitney U tests for continuous ones. Total T and B cell subset averages representing the means of CD31, CD41, CD51, CD81, CD101, and CD191 cells were calculated from two or three measurements performed in each heart sample. The cell populations present in each study group were compared using the t-test. A *p* value of <0.05 (two-tailed test) was statistically significant. All analyses were performed using R version 3.6 (R Core Team). The protocol of this study was approved by the research ethics committee of the Universidad Industrial de Santander CEINCI (Meeting No 11869 of 29 August 2014).

## 3. Results

A total of 32 cardiac explants were analyzed, 21 from patients with CCC and 11 from individuals with NCC, that had been collected between March 2014 and September 2017. Most patients were male (*n* = 26, 78.8%), with a mean age at the time of explant of 51.9 years (SD 2.1). No significant differences in the sociodemographic parameters were observed between the two groups. Similarly, the left ventricular ejection fraction value had a median of 11% (Q1: 10%; Q2: 15%) in both groups evaluated (*p* = 0.812) (Table 2). The most frequent diagnoses in the NCC group were dilated idiopathic cardiomyopathy (54.5%; *n* = 6), followed by hypertensive cardiomyopathy (18.2%; *n* = 2), ischemic cardiomyopathy (9.1%; *n* = 1), postinfectious cardiomyopathy (9.1%; *n* = 1), and valvular cardiomyopathy (9.1%; *n* = 1).

### 3.1. Macroscopic Analysis

Among the macroscopic characteristics of the explants from patients with CCC, the most notable was the presence of enlarged hearts, with a rounded and globular appearance secondary to marked myocardial flaccidity and the presence of large irregular pale areas suggestive of resolved episodes of pericarditis. In the comparative analysis, no differences were observed between the groups concerning organ weight, the presence of apical aneurysms, or intramural thrombi, as well as in the dimensions of the atrioventricular or semilunar valves (Table 2). However, we observed that the right ventricular walls were significantly less thick in explants from patients with CCC compared with those with NCC (0.58 cms vs. 0.75 cms, respectively. *p* = 0.005). Finally, no differences in left ventricular wall thickness were found between the two groups (Figure 1).

On the other hand, although no significant differences were observed concerning the presence of subendocardial fibrosis in the macroscopic analysis, the distribution of the same had notable differences between groups, a predominant localization towards perivascular areas was observed in patients with NCC, potentially associated with the presence of ischemic lesions secondary to atherosclerotic coronary artery disease. In contrast, none of the explants from patients with CCC showed any atheromatous involvement of the coronary arteries, and in six cases, we observed electrodes adhering to the wall of the right atrium. 

### 3.2. Microscopic Analysis

The microscopic evaluation of the different sections of cardiac tissue in the CCC group showed the presence of chronic active myocarditis, characterized by an abundant inflammatory infiltrate of mononuclear predominance with the presence of lymphocytes, plasma cells, histiocytes, and few eosinophils distributed in a nonuniform and diffuse manner between areas of fibrosis and areas with preserved muscle fibers (Figure 2A,C,E). In the comparative analysis, a significantly higher prevalence of diffuse inflammatory infiltrates as well as a higher degree of their severity was observed in explants with CCC in all segments evaluated (Table 2). In contrast, focal inflammatory infiltrates were more frequently observed in explants with NCC in left ventricular samples (Figure 2B and Figure 3).

Regarding evidence of fibrosis, a higher prevalence of interstitial fibrosis was present in explants with CCC in all segments evaluated except the right atrium, being of diffuse distribution (Figure 2A and Figure 3A). Similarly, the presence of perivascular fibrosis was identified as more frequent in the right ventricles of the hearts in the group with CCC (*p* = 0.031) (Figure 2C and Figure 3B). Likewise, foci of interstitial and perivascular fibrosis were observed in the tissues of the NCC group (Figure 2B,D) but when we compared both groups (CCC vs. NCC) the explants from the CCC group showed a higher frequency and extent of fibrosis in the right ventricular (*p* = 0.004) and interventricular septum (*p* = 0.011) sections (Figure 4A,B). Additionally, Masson’s Trichrome staining confirmed the presence of extensive areas of interstitial and perivascular fibrosis (Figure 5).

Microscopic analysis revealed the presence of necrosis alternated with findings of fibrosis and of changes in hypertrophy and elongation in the CCC group, whereas in the non-chagasic explants, necrotic fibers predominated in the foci of fibrosis (*p* < 0.001) (Figure 4C). Additionally, no foci or deposits of amyloid or iron were identified; however, a significant difference was identified in collagen deposits, being significantly more frequent in the right ventricle (*p* = 0.001) and interventricular septum (*p* = 0.007) samples of CCC patients (Figure 4D). To further characterize collagen’s role in CCC explants, immunohistochemistry staining was performed for type IV and XIV collagens. Type IV collagen was shown to be restricted to the basement membrane as well as being localized in the regions of replacement fibrosis and the interstitial matrix (Figure 6A). On the other hand, type XIV collagen was observed to be associated with fibrils in the stroma of the cardiomyocytes (Figure 6B). 

In addition, two main modifications of viable myocardial fibers were observed: hypertrophy and elongation (Figure 2E,F). These histopathological findings were present in both groups; however, important differences were observed in their distribution and severity, specifically a greater presence of elongated fibers in the CCC group in the septum (0.031) and right ventricle (*p* = 0.002) samples (Figure 4E). Finally, none of the samples evaluated from either group were *T. cruzi* amastigote nests detected, so PCR tests were performed in all the hearts of the CCC group to analyze the presence of the parasite. In these analyses, *T. cruzi* genome products were found in only 6 of the 21 hearts (28.6%). 

### 3.3. Cell Phenotyping

Finally, an analysis of the immunophenotype of the inflammatory cells in the cardiac tissue was performed in random samples from the two groups (four cases from the CCC group and five cases with idiopathic dilated cardiomyopathy from the NCC group). The absolute number of cells in the infiltrate was significantly higher in the group of explants from patients with CCC than in those with NCC, which demonstrated low cellularity. Additionally, more CD8+ T lymphocytes, memory CD8+CD45RO+ T cells, and memory CD4+CD45RO+ T cells were observed in the inflammatory infiltrate of patients with CCC than in those with NCC (*p* < 0.001, *p* = 0.02 and *p* = 0.009). In contrast, CD4+ lymphocytes and transition CD4+ and CD8+ cells (CD4+ CD45RA+/CD45RO+, and CD8+ CD45RA+/CD45RO+) were significantly more abundant in patients with NCC compared with those with CCC (Figure 7). No significant differences were observed in the other cell phenotypes. Finally, regarding regulatory T cells, an average of 2.2% FOXP3+CD25+ cells and an average of 9.4% CCR5+ cells were observed in the CCC group, whereas no such cells were identified in the NCC group. Therefore, we were unable to plot these comparative results in Figure 7.

## 4. Discussion

The results of the present study highlight the main histopathological features of cardiac tissue in explants from patients with end-stage chronic Chagas cardiomyopathy. These characteristics were compared with those of controls with non-chagasic heart failure and presented a higher prevalence of diffuse inflammatory infiltrates and a lower frequency of focal infiltrates in the group with Chagas disease. Similarly, the hearts of patients with CCC presented more frequent findings of global interstitial fibrosis, greater perivascular fibrosis in the right ventricle, and more degenerative myocytes and collagen deposits in the septum and right ventricle than in the hearts of the non-chagasic group. Finally, significant differences were observed in the cellular profiles of the inflammatory infiltrate between the two groups, specifically a higher proportion of memory T cells, mainly CD8+ T cells, and a lower proportion of transitioning T cells (CD45RA+/CD45RO+) in patients with CCC compared with the NCC group. The results of the present study allowed us to characterize the differential histopathological and phenotypic profile of CCC, describing in detail the unique findings of this severe cardiomyopathy. 

A relevant aspect of the present study corresponds to the similarity of conditions between the two groups, having comparable sociodemographic and left ventricular function characteristics. This was reflected in the similar macroscopic findings of the hearts of both groups, suggesting a similar stage of severity independent of the etiology. In contrast, it was at the microscopic analysis level that the most important differences between the two groups were observed, most notably a higher frequency and severity of inflammatory infiltrates in the CCC group. Specifically, the finding of a diffuse infiltrate profile in patients with CCC compared with a predominance of focal infiltrates in those with NCC reflects the severity of myocardial inflammatory involvement in Chagas disease. In contrast to ischemic cardiomyopathy, in which myocardial ischemia triggers an intense but mostly transient inflammatory response aimed at preparing the conditions for the proliferative phase of recovery of necrotic myocardium [12], Chagas cardiomyopathy is characterized by the persistent and chronic activation of the cellular immune response secondary to the persistence of *T. cruzi* antigens in the tissue and the dysregulation of compensatory anti-inflammatory mechanisms [13]. These processes explain the presence of a more abundant and diffuse inflammatory infiltrate in patients with CCC compared with their counterparts with other etiologies.

However, the mechanisms behind this imbalance are not yet fully understood. Specifically, it is characterized by a reduction in the anti-inflammatory responses, allowing the establishment of a pro-inflammatory profile, with fewer regulatory T cells, monocytes, and IL-10-producing T lymphocytes found in patients with CCC compared with those with the indeterminate form of the disease [14]. On the other hand, a significant increase in the number of TNF-alpha and IFN-gamma-producing cells both locally and systemically has been observed in CCC patients according to previous studies, specifically a predominance of CD8+ T lymphocytes in the myocardial tissue of patients with CCC, with their distribution being related to the areas with greater inflammatory involvement [15,16]. In particular, Fiuza et al. reported a significantly higher abundance of circulating memory T cells (IFN-γ producing CD8+ T cells) in patients with CCC compared with individuals with the indeterminate form of the disease, suggesting them as critical to the development of CCC via mechanisms related to IFN-γ. Likewise, an elevated percentage of central memory CD4+ T cells was observed in indeterminate patients after stimulation, suggesting an important role of these cell subtypes in modulating the host immune response potentially mediated by the increased control of cell migration to tissues [17]. Similar results were observed by Arguello et al., who assessed the expression of different markers for T and B cells in cardiac explants from patients with advanced CCC. In that study, the authors observed a high proportion of CD45RO+ T cells (predominantly CD8+ cells) with a Th1 profile and proliferative capacity infiltrating the myocardium of CCC patients [18]. These results resemble what we observed in our study, in which patients with CCC presented significantly higher percentages of CD8+, CD4+CD45RO+, and CD8+CD45RO+ T cells, showing abundant inflammatory infiltrates and extensive replacement of myocardial tissue by fibrosis. On the other hand, significantly more CD4+, CD4+CD45RA+/CD45RO+, and CD8+CD45RA+/CD45RO+ cells were observed in patients with NCC, which was associated with a milder inflammatory profile, with inflammatory infiltrates of lesser magnitude and more delimited distribution. Moreover, the low proportion of regulatory T cells observed in the CCC group may highlight the deficiency of IL-10-producing cells typical of the immunological imbalance leading to cardiomyopathy development [18,19]. In addition to partially validating the findings of previous reports, our study is to the best of our knowledge the largest comparing the characteristics of the inflammatory infiltrate in CCC patients with those of a control group with heart failure of similar severity but secondary to a nonchagasic etiology, providing key information about the specificity of these findings as a unique feature of the pathophysiology of CD.

Regarding the observation of type IV and XIV collagen expression in CCC samples, recent studies have indicated that nonfibrillar collagens (types IV, VI, and XIV) are also deposited during pathological wound healing and may play key roles in myofibroblast differentiation and the organization of the fibrillar collagen network [20]. Our results are in accordance with those of the study of Arciniegas et al., who observed more circulating anti-collagen type IV antibodies in symptomatic patients with Chagas disease than in asymptomatic individuals and healthy controls, which suggests the presence of an autoimmune response against structures of the viable myocardium in the course of CCC [21]. On the other hand, collagen XIV has been implicated as a regulator of fibrillogenesis due to its close interaction with the fibril surface. Previous studies have shown that this collagen type is present in tissues under high mechanical stress [22]. Nevertheless, collagen may also exert critical immunological functions, as it has been proposed that it may trigger the T cell exhaustion phenotype and also affect the functions and phenotypes of other immune cells such as tumor-associated macrophages in the setting of cancer [23]. 

Finally, an additional finding of our study corresponded to the low frequency with which it was possible to identify the parasite genome in samples of cardiac tissue from the left ventricle. Although previous literature agrees that in the chronic phase of the disease, the prevalence of the parasite in tissues decreases drastically compared with the acute phase, the frequency varies from one publication to another, with detection rates ranging from 57% to 100% [24,25,26]. However, in the present study, this value reached only 28.6%, with no direct correlation between the detection of the parasite genome and the severity of myocardial involvement, a finding that is still the subject of debate in the literature [15,25,27,28]. Despite this, our findings support the hypothesis of mechanisms in addition to direct parasite damage as primary contributors to myocardial damage and the progressive nature of the disease.

### Strengths and Limitations

One of the strengths of the present study is the possibility of performing a comparative analysis between patients with CCC and controls with heart failure of other etiologies but with similar sociodemographic characteristics and clinical disease severity. Moreover, this study is to the best of our knowledge the largest study performing this comparison by evaluating the composition of the inflammatory infiltrates in human myocardial tissue. 

However, we must also highlight multiple limitations of our study. First, the sample size evaluated was not large enough to identify significant differences in relatively infrequent macroscopic findings, such as ventricular aneurysms; therefore, we were able to identify only the largest statistical differences between the two groups in this aspect. Additionally, T cells were classified using mainly CD45 markers, but it was not possible to evaluate relevant markers such as CCR7. Similarly, it was not possible to characterize the profile of cytokines secreted by each T-cell type, which limited the possibility of correlating these findings with the pathophysiology of the disease.

## 5. Conclusions

The results of the present study highlight CCC as a cardiomyopathy with a unique form of myocardial involvement characterized by extensive and severe interstitial fibrosis, chronic inflammatory infiltrates involving both ventricles, collagen deposits, and the severe degeneration of the myocardial fibers. Interestingly, we observed a colocalization pattern between fibrosis and chronic inflammatory infiltrates, suggesting a cascade of events that include cell death by direct injury, the chronic release of proinflammatory cytokines, the massive recruitment of immune system cells, and the resulting extensive remodeling of myocardial tissue and its extracellular matrix, which ultimately leads to scarring and fibrosis. In addition, the predominance of memory CD4+ and CD8+ T cells in CCC compared with NCC hearts highlights the critical role of the parasite-specific lymphocytic response in the course of the infection and the unique pathophysiological mechanisms related to this immune response observed in CCC.

## Figures and Tables

**Figure 1 pathogens-11-01402-f001:**
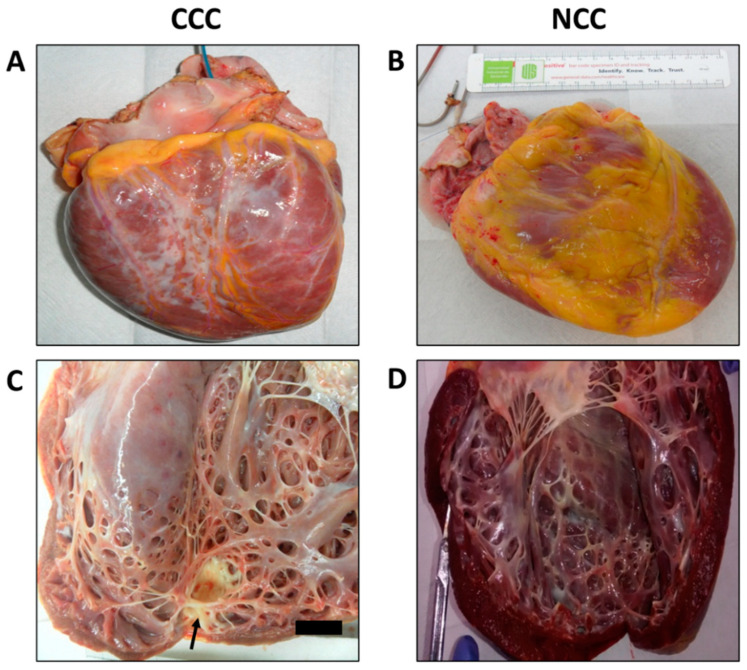
Macroscopic inspection of explanted hearts highlighting (**A**) globoid appearance of a heart affected by chronic Chagas cardiomyopathy compared with a (**B**) non-Chagas cardiomyopathy explant with a more preserved anatomical shape. (**C**) Highlights the marked dilatation and thinning of the left ventricular walls as well as the presence of extensive replacement of healthy myocardial tissue by fibrosis in the heart affected by CCC compared with the explant with NCC (**D**). The arrow in (**C**) indicates the presence of a left ventricular aneurysm, typical of advanced-stage CCC.

**Figure 2 pathogens-11-01402-f002:**
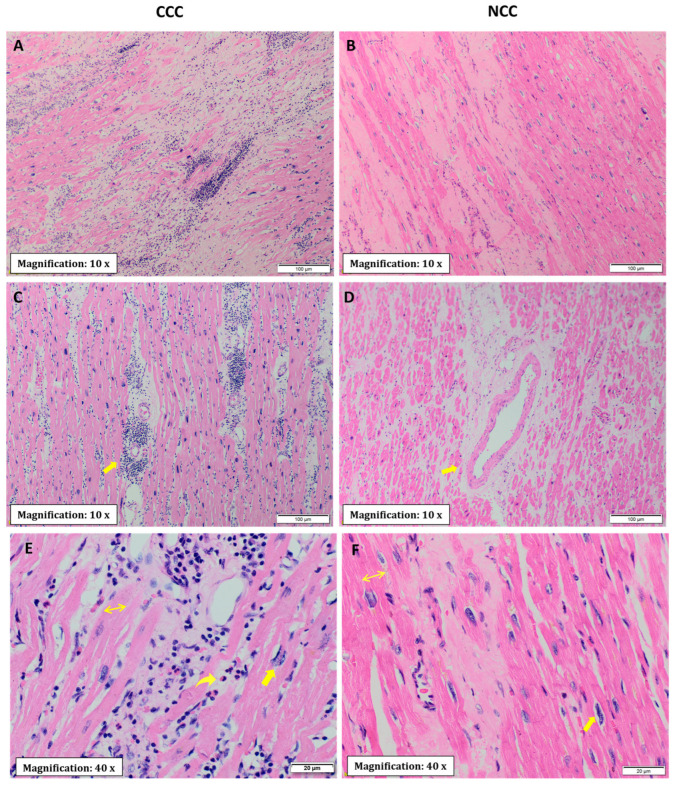
Histological section of myocardium stained with hematoxylin-eosin. The images in the left column (**A**,**C**,**E**) correspond to cardiac tissue from the CCC group, while those in the right column (**B**,**D**,**F**) correspond to the non-chagasic group. (**A**,**B**) Interstitial fibrosis, highlighting in the CCC group prominent lymphohistiocytic inflammatory infiltrate diffusely distributed. (**C**,**D**) The yellow arrows indicate perivascular fibrosis. Scale bars are 100 μm for (**A**–**D**). (**E**,**F**) Cardiomyocyte hypertrophy and areas of necrosis in both groups (CCC and NCC). The two-way horizontal arrow in both figures indicates increased fiber diameter. The unidirectional straight arrow in both figures indicates the presence of large, irregularly shaped nuclei. Finally, the presence of eosinophils in the CCC group is indicated by the curved arrow. Scale bars are 20 μm for (**E**,**F**).

**Figure 3 pathogens-11-01402-f003:**
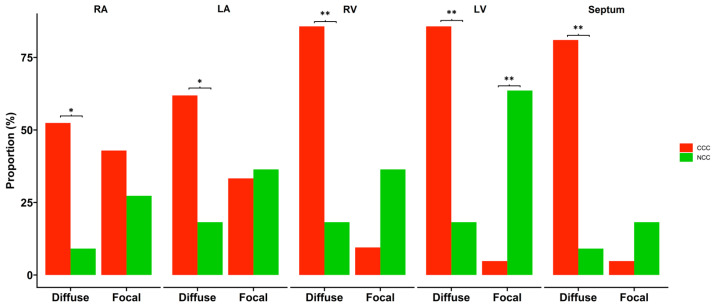
Comparison of inflammatory infiltrate type (focal vs. diffuse) according to the evaluated structure of patients with chronic Chagas cardiomyopathy (CCC) and non-Chagas cardiomyopathy (NCC). Bars represent the raw proportion of patients in which the mentioned alteration was present. Statistical significance: * *p* < 0.05; ** *p* ≤ 0.01. Abbreviations: RA: right atrium; LA: left atrium; RV: right ventricle; LV: left ventricle.

**Figure 4 pathogens-11-01402-f004:**
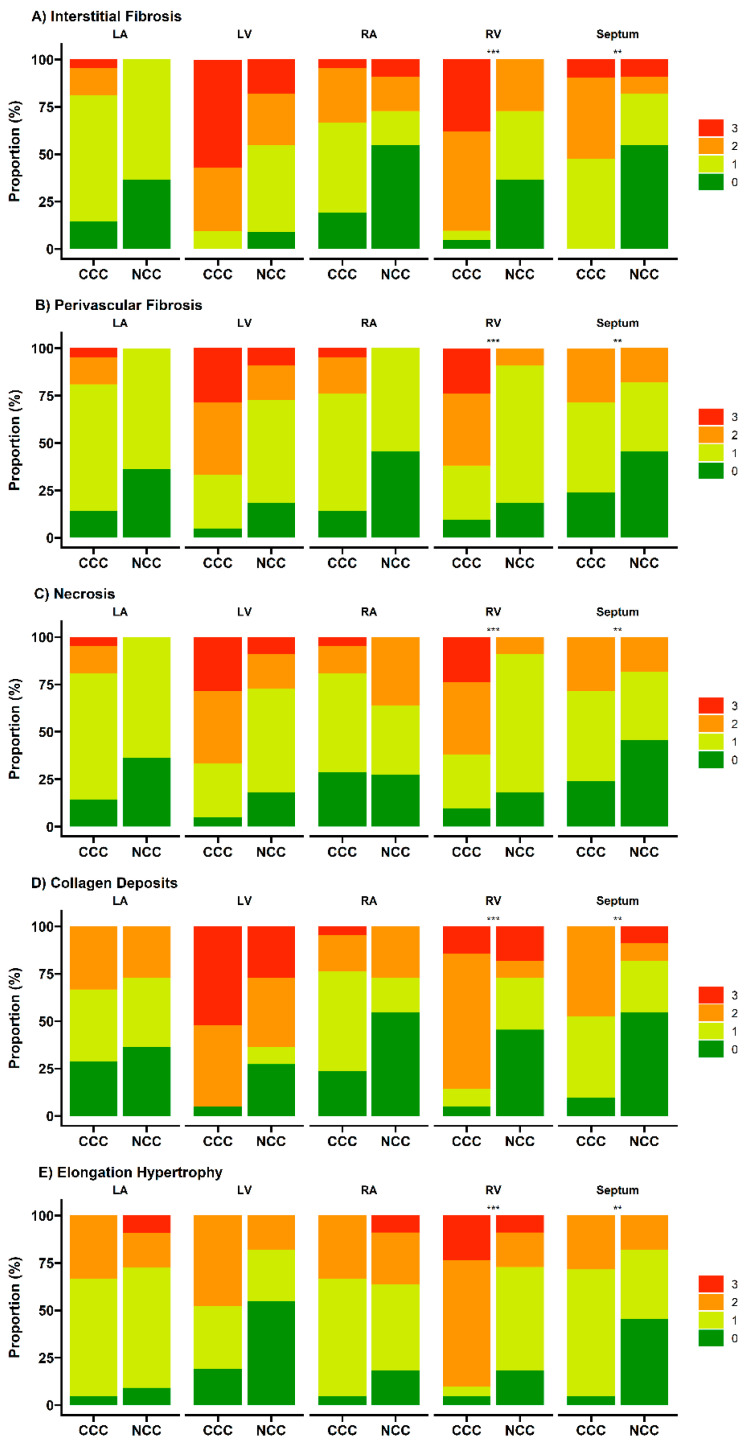
Histopathological findings in patients with chronic Chagas cardiomyopathy (CCC) and non-Chagas cardiomyopathy (NCC). The scale indicates the severity of the abnormal histopathological finding: 0: absent, 1: mild, 2: moderate, and 3: severe. Each bar represents the raw proportion of patients in each category. Statistical significance: ** *p* ≤ 0.01; *** *p* ≤ 0.001. Abbreviations: RA: right atrium. LA: left atrium. RV: right ventricle. RL: left ventricle.

**Figure 5 pathogens-11-01402-f005:**
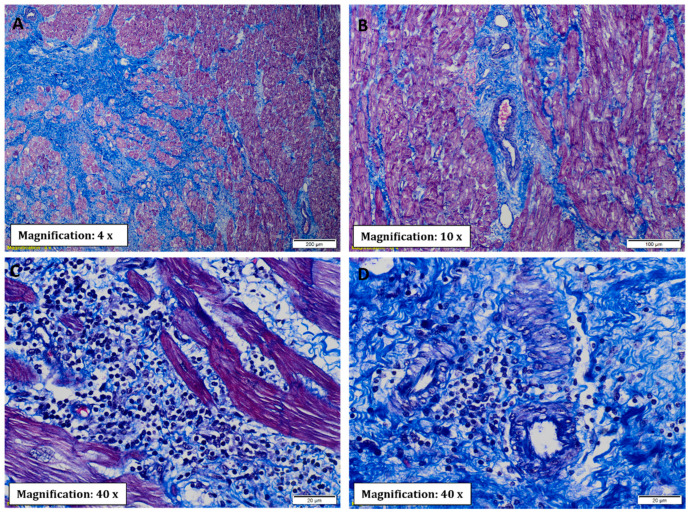
Masson’s trichrome stain showing extensive myocardial fibrosis in patients with CCC. (**A**) Extensive interstitial fibrosis represented by thick bands of fibroconnective tissue with extensive deposition of extracellular collagen matrix. The arrangement of the fibrous tissue separates the muscle fibers into small aggregates that give the appearance of islets. (**B**) Perivascular fibrosis. (**C**) Interstitial fibrosis in greater detail. (**D**) Colocalization of abundant inflammatory infiltrate and high degree of fibrosis with lymphocytes closely adhering to or in the vicinity of the sarcolemma.

**Figure 6 pathogens-11-01402-f006:**
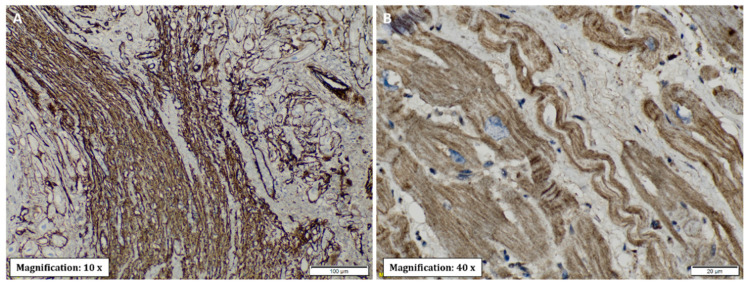
Type IV and XIV collagen immunohistochemical staining of cardiac tissue from patients with end-stage heart failure secondary to chronic Chagas cardiomyopathy. (**A**) Type IV collagen**.** Abundant immunoreactivity of type IV collagen, distributed in basement membranes and areas of interstitial fibrosis. (**B**) Type XIV collagen immunoreactivity shows cardiomyocytes with parallel fibrils directed longitudinally and the presence of a transverse pattern. The nuclei of the cardiomyocytes are shown in blue.

**Figure 7 pathogens-11-01402-f007:**
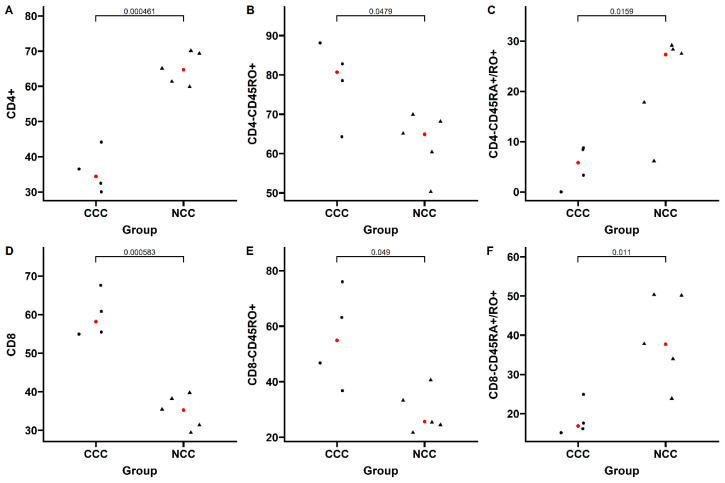
Immunophenotypic characterization of the T CD4+ and CD8+ cells composing the inflammatory infiltrate in explants with chronic Chagas cardiomyopathy (CCC) vs. non-Chagas cardiomyopathy (NCC). The median for each group is highlighted in red.

**Table 1 pathogens-11-01402-t001:** Panels for phenotypic characterization of inflammatory infiltrate cells by flow cytometry.

Panel	V450	V500	FITC	PE	PercPCy5.5	PeCy7	APC	APCH7	Cell Populations
Panel 1	CD20 (5 μL)	CD45 (3 μL)	CD3 (10 μL)	CD56 (10 μL)	CD4 (10 μL)	CD19 (5 μL)	CD8 (5 μL)	CD 14 (5 μL)	Lymphocytes
Panel 2	CD4 (5 μL)	CD45 (3 μL)	CD27 (10 μL)	CCR7 (10 μL)	CD3 (10 μL)	CD45RO (5 μL)	CD45RA (10 μL)	CD8 (5 μL)	T-cells subpopulations
Panel 3	-	CD45 (3 μL)	FOXp3 (5 μL)	CD25 (10)	CD4 (10 μL)	CD3 (3 μL)	CD27 (10 μL)	CD8 (5 μL)	Regulatory T cells
Panel 4	-	CD45 (3 μL)	FNTa (10 μL)	CCR5 (10 μL)	CD4 (10 μL)	CD3 (3 μL)	-	CD8 (5 μL)	Cytokines release

**Table 2 pathogens-11-01402-t002:** Sociodemographic, clinical and pathological characteristics of patients with chronic Chagas cardiomyopathy and non-Chagas cardiomyopathy controls.

Variables	Total (*n* = 32)	CCC (*n* = 21)	NCC (*n* = 11)	*p*-Value
Male sex	26 (81.3)	18 (85.7)	8 (72.7)	0.391
Age (years) ^•^	49.6 (13.2)	51.9 (2.1)	45.2 (5.5)	0.180
Ejection Fraction (%) ^‡^	11 (10–15)	12 (10–15)	10 (10–15)	0.815
Intramural thrombus (*n*,%)	3 (9.3)	2 (9.5)	1 (9.1)	0.999
Subendocardical fibrosis (*n*,%)	16 (50)	12 (57.1)	4 (36.4)	0.458
Aneurysm of the tip (*n*,%)	13 (40.6)	10 (47.6)	3 (27.3)	0.450
Weight of the heart (grams) *^,•^	487.5 (96.9)	485.2 (24.4)	492 (22.8)	0.862
Tricuspid valve circumference (cms) ^‡^	14 (13–14.5)	14 (13–14.5)	14 (13–15)	0.736
Pulmonary valve circumference (cms) ^‡^	9 (8–9.9)	9 (8–9)	10 (8.3–10)	0.161
Right ventricle wall thickness (cms) ^•^	0.65 (0.17)	0.58 (0.16)	0.75 (0.11)	0.006
Left ventricle wall thickness (cms) ^‡^	1.2 (1.1–1.25)	1.2 (1.1–1.2)	1.2 (1.1–1.5)	0.380
Mitral valve circumference (cms) ^‡^	11.4 (1.1)	11.3 (1.3)	11.5 (0.6)	0.627
Aortic valve circumference (cms) ^‡^	7.3 (0.6)	7.3 (0.7)	7.3 (0.6)	0.998

* *n* = 30, 1 patient in the group CCC > 600 g and one in the group CNC > 620; ^•^ Mean (Standard deviation); ^‡^ Median (Interquartile range).

## Data Availability

Data are contained within the article.

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
