# Peer review of "The Characterization of Cardiac Explants Reveals Unique Fibrosis Patterns and a Predominance of CD8+ T Cell Subpopulations in Patients with Chronic Chagas Cardiomyopathy"

_pathogens, 2022, doi:10.3390/pathogens11121402_

Round 1
Reviewer 1 Report
This is an important histologic and immunologic evaluation of Chronic Chagasic cardiomyopathy, highlighting the similarities and key differences in pathologic changes within the heart when compared to non-Chagasic inflammatory cardiomyopathies. Overall the report is very informative, but a few items require clarification as listed below:
1. In the abstract it is indicated that 32 explanted hearts were collected between 2014-2016, but in the Materials and Methods the collection dates are listed as 2014-2017 (Line 86). Please ensure the dates agree between the abstract and materials and methods.
2. In line 86, it is indicated that 21 explanted hearts were classified as CCC. How were they classified as CCC? Were the patients confirmed seropositive for T. cruzi on both a screening and confirmatory test? What is the interval between the positive test and the explant?
3. For phenotype evaluation of cells isolated from cardiac tissue (section 2.6.3), what concentration or absolute number of cells was used for each phenotyping panel?
4. Line 183, what version of FlowJo was used to analyze the flow cytometry data?
5. In table 2, please list the units of measurement used for each variable.
6. In figure 2, please include scale bars in panels B, C, and D. Also, please show all images at the same magnification for visual comparison of nuclear size, cardiomyocyte hypertrophy, etc. Also, it would be helpful to use arrows to emphasize areas of perivascular and interstitial fibrosis, bulky nuclei, different leukocyte types, etc.
7. For figure 3, please increase the font size of the axis labels as they are very small and difficult to interpret.
8. Figures 5 and 6, please include scale bars for all panels.
Reviewer 2 Report
The paper by Diaz et al. aimed to characterize the histopathological findings and the phenotype of inflammatory cells in the myocardial tissue of patients with end-stage heart failure (ESHF) secondary to CCC in comparison with ESHF secondary to non-Chagas Cardiomyopathies (NCC). The paper is well written and interesting. The study encompasses multiple techniques The authors conclude that: “our results contribute to a better understanding of CCC-specific immunoregulatory processes, potentially highlighting critical differences to other HF etiologies”
Although the question addressed in paper is of interest the paper itself suffers from methodological weaknesses and erroneous interpretations.
1 – The Figures must be improved since several one’s were presented without error bars. In general, the Figures must be improved. In addition, where are the FOXP3+CD25+ cells data referred on the results but never depicted in any Figure. In this regard, also, the author must quantify the results obtained in Figure 5-6.
2 – Finally, the authors pointed out that the results presented here “contributing to better understand of specific immunoregulatory processes”…Only a quantification of a few number of immune cells are not enough to define or to contribute to immunoregulatory process. In this line, the works appear to be more descriptive than mechanistic.
Round 2
Reviewer 2 Report
The authors improved the MS.